# MicroRNAs: Midfielders of Cardiac Health, Disease and Treatment

**DOI:** 10.3390/ijms242216207

**Published:** 2023-11-11

**Authors:** Emman Asjad, Halina Dobrzynski

**Affiliations:** 1Faculty of Biology, Medicine and Health, University of Manchester, Manchester M13 9PL, UK; emman.asjad@student.manchester.ac.uk; 2Department of Anatomy, Jagiellonian University Medical College, 31-034 Krakow, Poland

**Keywords:** miRNA, heart disease, biomarker, bradycardia, atrial fibrillation, heart failure, sinoatrial node, ion channels

## Abstract

MicroRNAs (miRNAs) are a class of small non-coding RNA molecules that play a role in post-transcriptional gene regulation. It is generally accepted that their main mechanism of action is the negative regulation of gene expression, through binding to specific regions in messenger RNA (mRNA) and repressing protein translation. By interrupting protein synthesis, miRNAs can effectively turn genes off and influence many basic processes in the body, such as developmental and apoptotic behaviours of cells and cardiac organogenesis. Their importance is highlighted by inhibiting or overexpressing certain miRNAs, which will be discussed in the context of coronary artery disease, atrial fibrillation, bradycardia, and heart failure. Dysregulated levels of miRNAs in the body can exacerbate or alleviate existing disease, and their omnipresence in the body makes them reliable as quantifiable markers of disease. This review aims to provide a summary of miRNAs as biomarkers and their interactions with targets that affect cardiac health, and intersperse it with current therapeutic knowledge. It intends to succinctly inform on these topics and guide readers toward more comprehensive works if they wish to explore further through a wide-ranging citation list.

## 1. Introduction

Structurally, miRNAs are small, non-coding RNAs which average around 22 nucleotides in length. Since their discovery in the nematode *C. elegans* in 1993 [1], over 2600 have been discovered in humans (according to the miRNA database) [2] and it is estimated they regulate one-third of the human genome [3]. Demonstrating this large involvement with the genome, they play vital roles in key biological processes, including cell proliferation, apoptosis, differentiation, and tumorigenesis [4]. This suggests that miRNA works in many different pathways, allowing more potential targets for miRNA therapies.

Proteins are regulated by miRNA acting on mRNA (messenger RNA) by binding to it post-transcriptionally. They can bind to different regions of target mRNAs, such as the 5′ or 3′ untranslated regions, which are binding sites for protein translation. They can also target the coding sequence or gene promoters (although most miRNAs interact with the 3′ untranslated region) [5]. This binding induces one of two things: mRNA degradation or translational repression. Degradation of mRNA takes place through a complex called “the RNA-inducing silencing complex” (RISC) cleaving the mRNA at a point complementary to the RISC. Conversely, translational repression occurs when this RISC blocks ribosomes from attaching to the mRNA sequence (thus preventing them from translating the sequence into amino acids).

Figure 1 shows the biogenesis of miRNA and protein synthesis. Canonical miRNA synthesis begins with miRNA translation from DNA sequences into long primary transcripts (pri-miRNA). This is usually via RNA polymerase II, although III is required for certain genes. The portion of the pri-miRNA encoding for miRNA then forms a hairpin loop structure that is recognised in the nucleus by a microprocessor complex. This complex is comprised of a ribonuclease III enzyme (DROSHA) and its co-factor DiGeorge Syndrome Critical Region 8 (DGCR8) and cleaves the pri-miRNA into precursor-miRNA (pre-miRNA). Once pre-miRNA is generated, it is moved out of the nucleus and into the cytoplasm by a protein complex (exportin-5/RanGTP [6]) to be processed by another ribonuclease III (DICER) into mature miRNA. This takes place through the removal of the terminal loop of the hairpin structure to form a double-stranded molecule with imperfect complementarity. One strand of this duplex becomes the guide strand and is loaded onto the Argonaute family of proteins, while the other strand (called the passenger strand) gets degraded. This combination of the guide strand with the Argonaute protein forms the RNA-Induced Silencing Complex (RISC) [7]. As the name suggests, these complexes negatively regulate mRNAs by silencing them. In disease, miRNAs can be both up- and downregulated, with a knock-on effect on mRNA expression.

When studying the effects of miRNA dysregulation, it is essential to distinguish between different mature miRNAs and gene loci, which is where the nomenclature comes into play. Families of miRNAs refer to groups with the same seed sequences, while clusters refer to miRNA genes physically close to each other on the genome (which can be from different families) [9]. A seed sequence is the 5′ prime end of a miRNA sequence, usually at nucleotides 2–7, and is an important determinant of target specificity and binding; several studies in animals have demonstrated that only the seed region is crucial for target recognition [10]. Furthermore, mature miRNAs are prefixed with ‘miR-’ and numbered in order of discovery. However, miRNAs followed by combinations of both numbers and letters are due to sequences varying by only one or two nucleotides [7]. This is exemplified in the sequence for miR-19a (UGUGCAAAUC**U**AUGCAAAACUGA) versus miR-19b (UGUGCAAAUC**C**AUGCAAAACUGA) [11]. Furthermore, miRNAs suffixed with ‘-5p’ or ‘-3p’ signify a 5′ or 3′ arm mature miRNA sequence, respectively [12]. 

According to the World Health Organisation, cardiovascular diseases are the leading cause of death globally, representing 32% of all deaths in 2019 [13], highlighting the importance of exploring new avenues of treatment. As already mentioned, this review will focus on miRNAs in healthy and diseased hearts, which are summarised in Table 1 below.

## 2. miRNAs in Healthy and Diseased Heart

### 2.1. miRNAs Use as Clinical Tools and Biomarkers

Genetic deletions of miRNAs in a range of organisms have been conducted to explore the importance of miRNAs; these reveal that few developmental processes completely depend on individual miRNAs. This indicates the presence of miRNA networks (rather than individual miRNAs) regulating distinct parts of pathways for a cumulative effect when it comes to regulating genes. It is suggested that this method evolved to counter deleterious variations within gene-expression programs [40]. Therefore, miRNAs can maintain normal cardiac function by acting as switches in overlapping networks that target mRNA expression. Based on data from a large sequencing project, the healthy adult heart expresses the following miRNAs, among others: miR-1, miR-16, miR-27b, miR-30d, miR-126, miR-133, miR-143, miR-208 and the let-7 family [41].

In particular, miR-1 is one of the most abundant miRNAs in heart tissue and is transcribed together with one of two 133-a genes as part of a bicistronic cluster (miR-1-1/133a-2 and miR-1-2/133a-1) [42]. These clusters play a role in the developing and postnatal heart, as shown by a study where targeted gene deletion of miR-1-2 in mice caused a spectrum of cardiac abnormalities that caused early lethality, cardiac rhythm disturbances in survivors, and myocyte cell-cycle abnormalities leading to hyperplasia of the heart [43]. This highlights how miR-1 has cardiac morphological, conduction, and cell-cycle limiting roles when expressed normally. Although its primary function relates mostly to cardiac development, it has postnatal effects on heart electrophysiology through physically binding with and suppressing an inward-rectifier potassium channel Kir2.1 [44]. In the primary pacemaker of the heart (sinoatrial node/SAN, which will be expanded on in detail in Section 2.2.2), miR-1 is responsible for the SAN development and regulates HCN4. This is a pacemaking channel that facilitates the funny current (I_f_), an essential ionic current within the SAN. There is evidence of decreased T-Box Transcription Factor 3 (Tbx3) in cells overexpressing miR-1. Important to the SAN development, Tbx3 down-regulates the atrial gene program as opposed to promoting SAN identity directly. miR-1 also causes a decrease in SAN-like cells through inhibiting mRNA translation rather than mRNA stability, leading to direct epigenetic modulation of the I_f_. This highlights the role of miR-1 in both cardiac development and heart rate modulation [45]. 

Further miRNAs of note are miR-208a/b and miR-499 which, alongside miR-1 and miR-133a, are cardiac-specific miRNAs (also known as myomiRs [46]). While miR-1/miR-133a play key roles in early development and electrophysiology, miR-208 and miR-499 are involved in late cardiogenic stages (as well as regulating sarcomeric contractile protein expression and expressing antiapoptotic roles respectively). Specifically, miR-208a/b is encoded within α-cardiac muscle myosin heavy chain gene introns (α-MHC, coded by *MYH6*) and β-cardiac myosin heavy chain gene introns, respectively (β-MHC, coded by *MYH7*) [47]. Although the role of miR-208 is not completely understood in the heart, luciferase reporter gene assays can be used to determine if it can activate or repress the expression of target genes. It does this by using a bioluminescent luciferase enzyme to highlight transcriptional activity [48]. Using this technique, *THRAP1* (now known as *MED13* [49]) is a direct target of miR-208, which represses its expression. In this way, miR-208 mimics could enhance cardiac function by repressing *THRAP1*, which increases β-MHC expression [50]. This increased expression is related to chronic cardiac stress in mouse models [51]. Protective miR-499 roles can be seen in ischaemic/reperfusion (IR) injury. This is demonstrated by increased miR-499 levels seen in rat models with IR injury, with evidence of it having cardioprotective effects against IR injury by targeting *PDCD4*. This is backed up by other evidence suggesting this mechanism may involve miR-499 inhibition of *PDCD4* and *PACS2*, which are both proapoptotic protein-coding genes, to prevent cardiomyocyte apoptosis [52]. It also highlights that individual miRNAs are as functionally important as transcription factors due to their regulatory effects [53].

There are two main approaches for miRNA-based therapeutics. One is restoring miRNA function via either synthetic double-stranded miRNAs or viral vector-based overexpression. The other is inhibiting miRNA function via chemically modified anti-miRNA oligonucleotides [54]. The various applications of these approaches are discussed further in this section and depicted in Figure 2. Oligonucleotides are synthetic DNA or RNA molecules that bind via Watson–Crick base pairing to either inhibit or increase the expression of its target RNA [55]; this is an essential base for many miRNA therapies. Reflecting this, anti-miRNA oligonucleotides (AMOs, or anti-miRs/antagomirs) inhibit endogenous miRNA by high-affinity binding which leads to inactivation and/or degradation of the target miRNA [56]. However, RNA nucleotide drug design is challenged by nuclease degradation upon introduction to biological systems, off-target effects, and activation of immune responses [57]. The latter is especially true for all miRNA therapies as miRNAs can have many targets in various complex pathways. This is exemplified in a study where an miR-34 mimic was administered to treat patients with advanced solid tumours, leading to adverse effects including fever and 4 immune-mediated patient deaths [58].

The degradation of miRNA therapies can be addressed by different delivery systems, but there are also more stable versions of AMOs with locked nucleic acid (LNA) nucleotides. LNAs have a modified ribose ring with an extra methylene group acting as a bridge between the 2′ oxygen and 4′ carbon of the ring; this structure increases resistance to enzymatic degradation, which is beneficial for drug delivery [61]. Other ways to inhibit miRNA include miRNA sponges, which use vectors with binding sites for multiple miRNAs so that they are less available to bind to their targets. Conversely, miRNA expression can be increased using miRNA mimics (double-stranded oligonucleotides) or vector delivery, such as through adeno-associated viruses (AAV). Certain AAV serotypes can target specific cell types without triggering inflammation [62], highlighting their potential as a delivery system that is both precise and able to avoid immune reactions.

The use of miRNAs against traditional biomarkers for different diseases is being explored, such as circulating troponins (cTns) for the diagnosis of acute myocardial infarction (MI). These cTns are considered a gold standard for early diagnosis of acute MI, and it is suggested they are released from necrotic myocardium before the onset of chest pain in acute MI. However, they are also raised in patients with end-stage renal disease, among other diseases. Studies testing the use of cardiac miRNAs against troponin T (cTnT) in patients with acute MI reveal that although all myomiRs (miR-1, -133a, -208b, and -499) are raised compared to healthy controls, they are not superior to the cTnT due to their differences in release during myocardial necrosis. Despite this, there may be potential in combining both markers for a more specific diagnosis which rules out non-cardiac events [63,64]. Therefore, the use of miRNAs as adjuncts to existing markers and techniques is promising. Further information on miRNAs as diagnostics in cardiovascular disease (CAD, acute coronary syndrome and HF) is compiled in distinct tables by Kramna et al. [65], while Sessa et al., convey a thorough systematic review of miRNA dysfunction studies, with a focus on the most studied miRNAs (miR-133-3p, miR-21, miR-499a-5p, miR-1, and miR-126) [66].

### 2.2. miRNAs in Disease 

#### 2.2.1. Their Role in Coronary Artery Disease

CAD is the most common type of cardiovascular disease, with the British Heart Foundation confirming it as the leading cause of death globally in 2019 [67]. It is estimated that globally, it affects approximately 1.72% of the world’s population and incidence continues to rise, with Eastern European countries experiencing the highest prevalence [68]. It is preceded by atherosclerosis (AS), which is a build-up of atheroma in the coronary arteries, causing vessel walls to narrow and harden. This makes it harder for oxygen-rich blood to be delivered to the heart and increases the probability of thrombosis and embolisms, which can lead to MI and stroke [69]. There are two theories for AS development: lipid infiltration and inflammation. The former involves low-density lipoproteins (LDL) and very-low-density lipoproteins (VLDL) undergoing oxidative modification and infiltrating the arterial wall. Macrophages then invade these lesions and engulf large amounts of lipids, becoming foam cells [70,71]. The inflammation theory builds on the inflammatory response. This response includes inflammation-activating endothelial cells, monocytes invading lesions, foam cells secreting inflammatory media, and macrophage apoptosis [70]. Both theories demonstrate immune system involvement, highlighting its importance in CAD (the various miRNAs affecting different parts of plaque build-up, such as inflammation, are shown in Figure 3 below).

In CAD, miR-126 is consistently seen to be downregulated when compared to non-CAD groups. It is abundantly expressed in endothelial cells and regulates components of the vascular endothelial factor (VEGF) pathway, which mediates angiogenesis [72]. This growth of new blood vessels from pre-existing vasculature can be a contradictory mechanism in CAD; on one hand, it is responsible for intraplaque haemorrhage, while on the other hand, angiogenesis therapy is being used to increase blood flow to ischaemic areas of the heart [73,74]. One role of miR-126 in VEGF pathways is its inhibition of *SPRED1* and *PIK3R2*, which are inhibitors of angiogenesis-promoting mitogen-activated protein kinase (MAPK) and phosphatidylinositol kinase (P13k) cell-signalling molecules respectively [75,76]. Furthermore, this miRNA also has anti-inflammatory effects by directly inhibiting vascular adhesion molecule 1 (VCAM-1) expression (which plays a key role in leukocyte trafficking to sites of inflammation), as well as inhibiting VCAM-1 upstream cell signals, such as tumour necrosis factor-alpha (TNF-α) [77]. The importance of this marker in the context of other diseases will be explored further later in this section.

**Figure 3 ijms-24-16207-f003:**
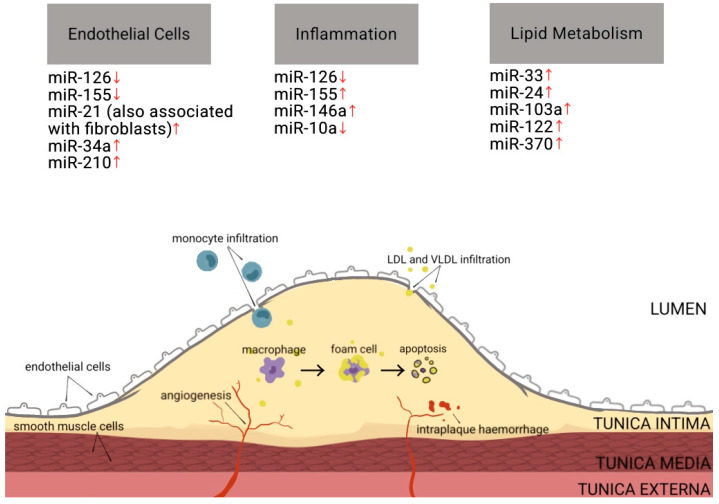
This is a diagram modified from Figure 3 in [78], depicting the various pathways of atherosclerotic lesion formation mentioned above. It also highlights some miRNAs altered in pathological processes involving endothelial cells, inflammation, and lipid metabolism within CAD. Note that some miRNAs are altered in more than one process. Arrows facing up depict upregulation while arrows facing down reflect downregulation in CAD. Three miRNAs (miR-126, -21 and -155) will be discussed in detail in the following section.

CAD also involves upregulated miRNAs such as miR-21, which has a positive correlation with disease severity, highlighting its capabilities of diagnosing CAD at different levels of progression. However, as it is not organ-specific, it would have to be combined with other miRNAs to be considered further [15]. It has beneficial roles, such as inhibiting *PTEN* (a tumour suppressor gene) and *PDCD4*, and thereby promoting cell survival and proliferation. However, it can also have detrimental effects, such as promoting cardiac fibrosis, which is a CAD complication. For instance, miR-21 increases transforming growth factor beta-1 (TGF-β1) by targeting *Smad7*, its inhibitory gene. TGF-β1 and miR-21 mutually increase each other’s production and are both seen to be upregulated in cardiac fibroblasts in the diseased state [79]. Another upregulated miRNA is miR-155, which is being evaluated for its therapeutic potential for targeting the macrophages that regulate inflammation. There are several activators for miR-155, including inflammatory stimuli such as TNF-α (which is part of a positive feedback loop with miR-155) and oxidised LDL. Inhibition of this miRNA using AMOs shows decreased pro-inflammatory cytokines and a shift from M1 macrophages, such as TNF-α and IL-12, to M2 macrophages (which promote cell proliferation and repair), such as IL-10. Inhibition of miR-155 also demonstrates increased IL-13, which regulates genes that are essential for M2 phenotype development. This is further evidence of the role of inflammation and immune-driven miRNA involvement in CAD, as macrophage infiltration can promote the growth of plaques. By repressing or increasing certain miRNAs, we can induce AS regression [16]. In relation to CAD, Ghafouri-Fard et al. reinforce miRNA roles in vascular smooth muscle cells, inflammation, and angiogenesis. They also address miRNAs targeting many parts of the same pathways and miRNA age-related changes [80].

Patients with type 2 diabetes (T2D) are twice as likely to develop CAD compared to healthy individuals, and there are 39 overlapping miRs that could aid in early disease detection as miRNAs have the advantage of highlighting genetic changes long before disease onset. For instance, increased expression of miR-126 achieved a satisfactory overall predictive ability in diagnosing T2D, though the value was significantly lower than that calculated for glycated haemoglobin (HbA1c, which is a common diagnostic marker for T2D). However, HbA1c is only useful when diabetes is already established, so further experiments for early diagnosis using miRNAs still have potential. A positive correlation between high-density lipoprotein (HDL-C) and miR-126 can also be seen, where individuals with diabetes who have both low miR-126 and low HDL-C have a higher risk of comorbid CAD. It is essential to consider the “common soil” hypothesis, which refers to how certain diseases (such as CAD and T2D) can have similar pathogenic backgrounds and marker profiles [14], so we may need to combine miRNAs with each other (or with other markers, such as HDL-C) to ensure they are differential. Furthermore, many biomarkers can be used to differentiate between CAD and non-CAD patients (such as miR-765, -149, -424, and -133), with miR-133a exceeding the prediction potentials of demographical data and risk factors, while miR-181 is also associated with CAD (even after adjustment for traditional risk factors, such as obesity) [17]. There are many risk factors to adjust for when considering the clinical application of miRNAs, such as hypertension, lipid imbalance, and cigarette smoking. These can all contribute to metabolic syndrome, which activates plaque formation and promotes thrombosis and inflammation [81]. Therefore, it is important to account for risk factors when exploring the potential of new biomarkers to see how versatile they are in different contexts of the same disease.

#### 2.2.2. Their Role in Atrial Fibrillation and Bradycardia

AF is the most common sustained cardiac arrhythmia. It is especially seen in the older population, which is growing due to advances in the management of underlying conditions. In the United Kingdom, its cost burden is predicted to increase from 0.9% to a possible 4.27% of the National Health Service’s (NHS) expenditure in the next two decades [82]. The Global Burden of Disease study in 2019 shows a ~1.4-fold increase in AF deaths globally from 1990 to 2019 [83]. Electrical conduction is abnormal in AF, leading to an irregular heart rhythm that causes the atria to fibrillate/quiver. This leads to turbulent blood flow, increasing the chance of clots [84] and predisposing AF patients to conditions such as HF and ischaemic stroke. Although the exact process is unknown, there are three potential theories to explain its pathophysiology: the focal mechanism, the single circuit re-entry, and the multiple wavelet theories. The first theory suggests impulse firing from areas of the heart outside of the normal pacemaker cells [85]. The second theory suggests that loops of self-sustaining electrical activity interfere with the natural pace-making of the heart [86]. The final, most widely held theory suggests multiple random, wandering areas of disorganised electrical signals that can collide and interact with one another and perpetuate the arrhythmia [87].

In AF, miR-1 upregulation can be noted. Overexpression of miR-1 slows cardiac conduction, as evidenced by QRS complex widening and prolonged QT intervals [18]. This miRNA can be inhibited by metformin, which is typically used to treat T2D patients. Metformin inhibits miR-1 by acting as an allosteric activator of adenosine monophosphate-activated protein kinase (AMPK). This protein kinase negatively regulates the transcription factor CCAAT/enhancer-binding protein beta (C/EBPβ), which directly promotes miR-1 [88]. Decreasing miR-1 levels improves arrhythmogenic symptoms as it represses cardiac excitability genes such as *GJA1*, which codes for connexin43 (Cx43). Cx43 is a cardiac gap junction channel responsible for impulse conduction between cells by allowing cell-to-cell movement of ions [89]. *KCNJ2* is another such gene, which encodes the Kir2.1 protein that mediates potassium selective inward rectifier current (I_K1_) [18]; this channel controls the resting membrane potential of a cell and contributes to the repolarising current at the end of an action potential [90], making it important for regulating action potential duration (APD). Downregulation of miR-1 is therefore beneficial in the context of AF, and high levels have the potential to be treated by an existing drug. This idea of using already-developed drugs could also be applied to other miRNAs in other contexts.

One characteristic of AF is its persistence, even in the absence of underlying disease, due to how it leads to atrial remodelling and perpetuates itself. This remodelling process is closely linked to a significant reduction in L-type Ca^2+^ current (I_CaL_) and a shortening of atrial APD to avoid Ca^2+^ overload during the increased firing rate that is seen in AF. It is therefore important to understand that calcium is a vital contributor to cardiac contractility. In particular, miR-328 is seen to be elevated four times in AF over controls. It is linked to this remodelling process as it potentially targets *CACNA1C* and *CACNB1*, which encode L-type Ca^2+^ channel subunits in the heart. A study involving adenoviral overexpression of miR-328 in dogs resulted in decreased I_CaL_, shortening of APD, and enhanced susceptibility to AF. These effects were consequently able to be reversed by antagomir-328. While miR-328 is shown to be high in left atrial tissue specifically [19], other AF studies have found it to be comparatively lower in plasma [20,21]. This highlights the differing concentrations of certain miRNAs, even in different parts of the same organ. Ravelli et al. efficiently guide attention toward studies which show both direct and indirect roles of miRNAs in AF, with indirect pathways including their contribution toward oxidative stress and inflammation processes rather than ion channel remodelling directly [91].

The SAN is the primary pacemaker of the heart and, like other heart structures, is influenced by miRNAs. It is a part of the cardiac conduction system (CCS) and is made up of specialized cardiomyocytes distinct from the working myocardium. To facilitate action potential generation and propagation, it possesses a unique set of ion channels. As already stated, the “funny current” (I_f_) is a well-known ionic current in the SAN. It is facilitated by ionic channels from the *HCN* gene family. Of the four isoforms within this family, the I_f_ expresses intermediate characteristics between *HCN1* and *HCN4*. These two isoforms are abundant in the SAN and mainly absent in the atrial muscle [92], highlighting the I_f_ as a target for chronotropic drugs. It is a mixed Na^+^/K^+^ inward current, activated upon hyperpolarisation at around −45 mV and reversed at approximately −10/−20 mV; indeed, the funny current is named as such due to these unusual properties [93]. Key miRNAs which impact *HCN4* channel expression, and the subsequent effects, are conveyed in Figure 4 below. The SAN also expresses Ca^2+^-handling proteins, which, alongside the ion channels, are responsible for the two main mechanisms responsible for SAN automaticity. This coupled-clock system includes the membrane voltage clock (M clock) and the Ca^2+^ clock [94]. The former refers to plasma membrane ion channels carrying an inward current while other ion channels carry an opposite and outward current; the latter refers to intracellular signals, such as local spontaneous Ca^2+^ rhythms present in the sarcoplasmic reticulum, driving the pacemaker potential [95].

Awareness of how genes and molecules in the SAN are expressed differently from the atrial muscle is important when exploring the impact of miRNAs on pacemaking. For instance, there are several miRNAs expressed at lower levels in the SAN (versus the atrial muscle), which downregulate pacemaking mRNAs known to be expressed at higher levels in the SAN. These include miR-486-3p, which is confirmed to target mainly *HCN* channels [31]. To delve further, miR-1/-133 can once again be discussed. Consistent with their role in early cardiac development, both have been found in neonatal hearts, with even higher levels being maintained in the adult heart. This enhanced expression of miR-1/-133 in adults has been seen to limit *HCN2/HCN4* genes. *HCN2* and *HCN4* have been seen to increase in failing hearts, contributing to arrhythmogenesis, and when miR-1/-133 levels drop in instances such as cardiac hypertrophy, these genes can re-express and negatively modulate pacemaking [34]. This concept is seen in a study where right atrial appendage samples were collected from patients undergoing coronary artery bypass grafting and were compared to older patients with sinus rhythm; those with AF demonstrated significantly higher *HCN2/HCN4* levels alongside significantly lower miR-1/-133 levels [97]. This highlights an important link between the SAN and ageing; if age is a factor that correlates with levels of certain miRNAs which can be measured, then illness could be detected early.

Through poorly understood mechanisms, SND and AF are conditions that create a foundation for each other to manifest and perpetuate. Together, AF and SND form the basis of “tachycardia-bradycardia syndrome” (a variant of SND in which slow and fast arrhythmias alternate [98]), highlighting how an amalgamation of AF with other conditions can go further and beget many different types of pacemaking diseases. Specifically, sick sinus syndrome (SSS) was coined in 1968 to group all the anatomic aetiologies of SND [99]; it is the most frequent indication for electronic pacemaker implantation, making it an important clinical concern. In relation to SSS, miR-17-92 and miR-106b-25 are seen to be positively regulated by transcription factor Pitx2 (Paired Like Homeodomain 2). This directly inhibits transcription factors Shox2 (SHOX Homeobox 2) and Tbx3, which promote differentiation into SAN cells [35]. Though Shox2 is required for SAN differentiation in mice, it has not been implicated in human cardiac dysfunction yet; conversely, Tbx3 has been implicated in human PR interval prolongation and has been seen to induce pacemaker properties in the adult myocardium when incorrectly expressed [100]. This could suggest a cause for pacemaker dysfunction and an origin for ectopic signal firing which can contribute to AF. It also highlights a possible avenue of repressing specific genes which modulate miRNAs, rather than miRNAs directly (especially when there are several miRNA targets).

Another miRNA of note is miR-1976; this is upregulated in age-related SSS patient plasma compared to controls, which is confirmed by miRNA profiling via real-time quantitative reverse transcription PCR (qRT-PCR, a technique used for qualitative analysis of gene expression [101]). Age-dependent decreases in Ca_v_1.2 and Ca_v_1.3 calcium channels play a role in SAN degeneration, and miR-1976 targets both. This is highlighted in homozygous mice, where miR-1976 levels were >2-fold higher than in homozygous or wild-type mice. Alongside lower intrinsic heart rate in homozygous versus wild-type, Ca_v_1.2 and Ca_v_1.3 protein levels were also significantly decreased. This shows a clear target of miR-1976 and a potential non-invasive therapy for age-related SSS [36], especially in patients who already have AF (it would be useful to prevent SSS in people where it would do more damage). Furthermore, miR-423-5p can also be implicated in bradycardic arrhythmia onset, as it downregulates the *HCN4* pacemaking channel. Bradycardia is another type of arrhythmia, characterised by a heart rate lower than 60 beats per minute [102]. miR-423-5p is seen to be increased by exercise training, which then induces *HCN* remodelling. This relationship was tested by injections of cholesterol-conjugated anti-miR-423-5p in sedentary and trained mice, which resulted in abolishing/blunting of the training-induced bradycardia in the trained mice while having no effect on heart rate for the sedentary mice. This relationship was also true of humans, where ivabradine (blocks *HCN4* and I_f_) was administered to athletes and non-athletes. Those with lower heart rates (generally athletes) had a blunted response to the drug, suggesting the downregulation of *HCN4* and I_f_ [37]. This suggests a possibility of lifestyle/exercise adjustments to treat certain conditions, with a measurable variable (miRNAs) to monitor how well these adjustments may be working.

Currently, there is no specific treatment for AF that decreases AF mortality other than anticoagulants [103]. Altogether, 285 miRNAs are seen to be dysregulated in several studies regarding AF. Of those, 69 have been validated using qRT-PCR, and modulation has been shown to protect from AF in animal models in only six (miR-1, -21, -26, -29b, -31 and -328). Although caution should be taken when extrapolating results from animal models [22], there is promising evidence for those that have been whittled down, of which four have already been discussed in this report (including miR-21, which exhibits promotion of cardiac fibrosis in AF as well as CAD [104]). Cardiac fibrosis in atrial remodelling is also seen as a result of downregulated miR-29b. This is evidenced in a study where miR-29b sponge-bearing viruses were injected into animal models, resulting in decreased fibroblast miR-29b by 68% [23]. Fibroblasts are cells responsible for the deposition of extracellular matrix (ECM) which provides structural support in the heart; in pathological remodelling, fibroblast activation and proliferation can contribute to cardiac fibrosis [105,106]. However, despite this combined research, there is no current, developed clinical application for miRNAs in AF beyond their role as possible biomarkers.

#### 2.2.3. Their Role in Heart Failure

Cardiovascular diseases “often result in a phenotypically similar endpoint” which is HF, due to stressors that lead to maladaptive cardiac recovery and remodelling [62]. According to the National Institute for Health and Care Excellence (NICE), HF produces raised intracardiac pressures and/or inadequate cardiac output, as a result of functional or structural abnormalities [107]. These abnormalities can take place through fibroblasts. Inflammation and impaired vascularisation can also contribute to HF, demonstrating further how non-cardiomyocytes (such as fibroblasts and endothelial cells) can promote deteriorating heart function [108]. However, these factors depend on the initial pathology causing the HF. This highlights why HF is a focus in itself; any heart disease can function as an index event for its pathogenesis [109]. Reflecting this, the following section will include previously mentioned miRNAs and cellular pathway components, while interspersing them with new information specific to HF.

Many miRNAs previously mentioned in this review can be seen to be modified within HF. For instance, miR-21 upregulation leads to pro-fibrotic pathways; miR-126 downregulation leads to loss of vascular integrity; miR-1 electrophysical alterations cause changes through ion channels [109]. Having diagnostic combinations of miRNAs in the future may be useful to identify which miRNAs converged to trigger HF, and therefore, what the preceding cause was. One set of miRNAs that can be discussed further is the miR-29 family. In the context of HF, they are downregulated and target multiple fibrosis-related genes, such as *COL1A1*, *COL1A2*, *COL3A1*, *FBN1*, *IGF-1*, and *PTX3*, among others [110,111]. Although the exact mechanism is unclear, *TGF-β1* is a pro-fibrotic gene that can activate its downstream mediator *Smad3* to inhibit miR-29, resulting in increased fibrosis. This effect is seen in mice lacking *Smad3*, which are protected against fibrosis in many disease models, including ischaemic and hypertensive cardiac remodelling. Although this inhibitory miR-29 mechanism is not limited to the heart [24], the concept could be applied to future therapies as a preventative measure to protect individuals from fibrosis-related HF. 

There are also miRNAs involved in cardiac hypertrophy, which is associated with a significantly increased risk of HF [112]. One such miRNA is miR-200a-3p, which directly targets both *WDR1* and *PTEN* and acts as a positive modulator of cardiac hypertrophy. *WDR1* is involved in myocardium sarcomere organisation, and its deletion in mice models leads to cardiac hypertrophy, electrocardiogram abnormalities, and early death. Suppression of *PTEN* can also be detrimental, as it is a negative modulator of the PI3K/AKT pathway, which is known to be involved in hypertrophy onset [25,26]. Therefore, inhibition of miR-200a-3p to target several mediators of cardiac hypertrophy may play a role in preventing HF. Furthermore, miR-378 is a muscle-enriched miRNA expressed in cardiomyocytes but not fibroblasts. Maintenance of its levels via genetic overexpression can decrease cardiac hypertrophy [27]. There are several factors in the MAPK pathway known to be involved in cardiac hypertrophy, such as *IGF-1*, *KSR1*, and *MAPK1*. There is evidence of miR-378 directly repressing *MAPK1* which, alongside *MAPK3*, acts as a signal transductor that integrates upstream signals and converts them into fewer, uniform responses, such as proliferation, apoptosis inhibition, and cell growth. These are fundamental biological processes, and therefore, the role of miR-378 in regulating them is significant. As miR-378 is seen to be significantly downregulated in cardiac disease [113], it may be a good therapeutic target for cardiomyocyte dysfunction specifically (and therefore aid in preventing hypertrophy-related HF).

Pro-B-type natriuretic peptide (pro-BNP) is a hormone, which upon release into circulation, cleaves into an inactive peptide N-terminal-pro B-type natriuretic peptide (NT-proBNP) and active B-type natriuretic peptide (BNP). This takes place in response to myocardial wall stress, such as in HF [114]. These natriuretic peptides are gold standard biomarkers for HF diagnosis; BNP production in healthy people is usually low, at around 10 pg/mL [115]. In a study of HF patients against controls, miR-423-5p is seen to be elevated, with its levels relating to NT-proBNP. To rule out miRNA changes due to dyspnoea (breathlessness), non-HF patients with this symptom were also included, highlighting that miR-423-5p had a differential potential for HF specifically. It was seen to be a significant predictor for diagnosis even when accounting for age and sex, showing a three-fold increase in failing myocardium as compared to healthy human hearts [28]. Indeed, Gomes et al. address the exacerbating/alleviating effects of miRNAs in HF (regarding hypertrophy, inflammation, regeneration and vascular remodelling) through highlighting mechanisms of action [116]. Being able to compare similar gene targets and mechanisms of actions between miRNAs opens up further points of intervention when developing cardiac therapies in the future.

HF, AF, and SAN dysfunction are conditions that frequently co-exist. HF can give rise to AF via atrial remodelling, occurring through triggers that include abnormal intracellular calcium handling and atrial stretch. Crosstalk between ectopic electrical activity and the SAN can, in the presence of HF, critically modulate the development of AF and SSS [117]. Furthermore, SAN dysregulation in response to stressors, such as exercise, can lead to exercise intolerance in HF patients. This is seen in rat models of HF with preserved ejection fraction (HFpEF, patients with >50% left ventricular ejection fraction [118]), where this chronotropic incompetence presented alongside an alternating leading pacemaker within the SAN. Single-cell studies also reveal coupled-clock alterations as a result of HFpEF [119]. This highlights how adjusting lifestyle factors, such as increasing exercise, may not always be feasible for HF patients looking to prevent further issues. Here, miRNAs may be effective; mouse models of HF with sinus bradycardia show upregulation of miR-370-3p, which is seen to downregulate *HCN4* channels. However, intraperitoneal injection of anti-miR-370-3p into HF mice restores *HCN4* mRNA and I_f_ while also partially restoring ventricular function and reducing mortality [38]. Of note, *KCNA4* (encoding a K_v_1.4 current essential for APD) expression is seen to be diminished in ischaemia and HF human tissue. It is targeted by miR-448, and the inhibition of this miR restores *KCNA4* and has antiarrhythmic effects [39]. In this way, restoring enough function to ensure a patient can try to stay fit and healthy is a promising route to explore.

Significant upregulation of several miRs can be observed in HF SAN overall (“miR-3200-3p, let-7g-3p, miR-486-3p, miR-652-5p, miR-133a-3p, miR-1-3p, mir-30c-5p, and miR-187-3p”). These are linked to downregulating expression of key ion channels for pacemaking and conduction, including *HCN1/4* (I_f_), alongside *SCN1A*, *SCN8A*, *SCN1B* and *SCN2B* (responsible for I_Na_). Supporting this, miR-486-3p is confirmed to depress SAN automaticity and *HCN4* expression [32], and significantly down-regulates hsa-Cav1.3 and hsa-Cav3.1 calcium channels, observed through reduced channel luciferase activity. Conversely, miR-486-3p is seen to alleviate the bradycardia that is specifically seen in COVID-19 patients. This highlights the immune system’s role in HF; through its inhibition, a tryptase-mediated hypertrophy pathway is suppressed [33]. There are also significantly downregulated miRs (“let-7a-5p, miR-1247-3p, miR-423-5p, mir-574-5p, and mir-25-5p”), which are linked to downregulating potassium channels vital for both cardiomyocyte repolarisation and SAN automaticity. These are *KCNJ3/KCNJ5* genes (coding GIRK1/GIRK4, G protein-coupled inwardly rectifying potassium channels), and *KCNQ1/KCNE1* genes (coding IKs, delayed rectifier potassium channels) [32]. Notably, it is speculated that, alongside miR-29c-3p and miR-584-5p, miR-1247-5p regulates tumour suppressor genes, such as *p53* and *TGF-β* signalling pathways, and thereby influences structural remodelling in the heart [120]. This is a possible complication as modulating miRs which impact apoptotic genes such as these is not specific and could cause adverse effects.

In terms of using miRNAs as prognostic tools, this can be seen in patients admitted to the hospital with acute decompensation (sudden or gradual onset of HF symptoms requiring unplanned medical involvement [121]). Patients had increased levels of miRs-21, -126 and -423-5p between admission and discharge, and decreased levels following clinical decompensation. Those with increased levels of miRs-21 and -126 at the time of treatment exhibited better 24-month survival and longer rehospitalisation-free periods, while increased miR-423-5p also meant fewer hospital admissions in the 24 months following (compared to patients with decreased levels) [29]. This demonstrates how the same miRNA could act both diagnostically and prognostically in future clinical applications. To add, in the first clinical trial of antisense drug administration in HF patients, miR-132-3p was inhibited by antisense oligonucleotide CDR132L. This miR is upregulated in HF patients and inhibits anti-hypertrophic, calcium handling, and contractility genes, driving adverse cardiac remodelling. CDR132L demonstrated potent target reduction in plasma, along with the reduction of NT-proBNP and QRS complex narrowing. Though this was a small study, treatment was well-tolerated, demonstrating promise in targeting miRNAs for future treatments [30]. Aside from CDR132L, 10 other miRNA-based clinical trials have been elegantly reviewed by Laggerbauer et al. [5].

## 3. Future Directions

Future research is encouraging for miRNAs, especially as biomarkers. There are many strengths to this approach, including miRNA cell specificity, their role as sensitive indicators of change in response to toxicants/infection/stress and extracellular release, and their ability to form complexes to become stable and avoid extracellular RNase degradation. Specifically, their ability to be measured non-invasively is a crucial factor to consider against other biomarkers. However, these upsides are challenged by considering sample types and purification methods, which can affect miRNA levels. This stresses the need for a standardized processing protocol for accurate results [122]. Nevertheless, there are many instances of miRNAs used as biomarkers.

There are examples of miRNAs as diagnostic tools; this is highlighted by miR-499, which exhibits an >80-fold increase in acute non-ST elevation MI compared to controls and has higher diagnostic accuracy than cTnT [123]. One further application could be the diagnosis of transplant rejection, explored by a study where heart transplant patients with no history of rejection were compared with those who did. Plasma samples underwent small RNA sequencing to reveal several miRNAs that discriminated between two subtypes of rejection. This reveals a noninvasive way to diagnose organ rejection as opposed to biopsies [124]. Interestingly, miRNAs may also be able to facilitate sex-specific diagnoses, such as in unrecognised HFpEF following diabetes. This occurs more commonly in women than men, and low levels of miR-34a, -224, and -452 are seen to be specifically targeted to women with HFpEF [125]. Additionally, miR-133a has been seen to correlate with all prognostically relevant cardiovascular magnetic resonance imaging for MI damage, including infarct size and salvaged area at risk. However, it does not add individual value to traditional markers of prognosis, and more research is needed here [126]. They also have potential as predictors of disease, demonstrated by significantly low serum circulating miR-150 levels in patients with post-acute MI HF (as opposed to those with post-acute MI non-HF). Furthermore, miR-150 showed increased predictive accuracy for post-acute MI HF when compared with traditional HF biomarker BNP; notably, combining both was an even more powerful predictor. However, a larger study is needed to validate these predictive qualities [127]. Predisposition to certain conditions can also be affirmed by miRNAs, such as in Takotsubo syndrome (TTS), an acute form of HF triggered by high adrenaline during stress. Circulating miRs can be used to both discriminate between TTS and MI and highlight their involvement in manifestation in each condition. This is seen in how AAV co-expressing miR-16 and miR-26a yield TTS-like changes following adrenaline administration in vivo [128]. Discriminative qualities of miRs are further emphasised by miR-29-5p, which presents at higher levels in serum in HF patients with idiopathic dilated cardiomyopathy (left- or bi-ventricular dilatation and impaired myocardial contractility due to an unknown cause [129]) compared with ischaemic heart disease (CAD) [130]. Altogether, the ability of miRNAs to diagnose, discriminate, predict, and support prognoses for diseases, as well as outperform certain traditional biomarkers, makes them multi-faceted and incredibly diverse molecules despite certain areas needing more research/larger study samples.

## 4. Conclusions

Due to their widespread existence in the body, miRNAs are being extensively researched, not only to highlight their roles throughout the body but also as treatments and biomarkers. Their ability to correlate with traditional biomarkers gives miRNAs potential. Despite the current lack of standardization, among other challenges, it has been repeatedly highlighted throughout this article that combining miRNAs, both with each other and with other biomarkers, shows potential. miRNAs are the go-between for most biological processes, and the ability of these ‘midfielders’ to both precisely target certain genes and be part of extensive gene networks means we must be careful of serious adverse effects when progressing to clinical use, as shown by the overlap of gene targets and pathways throughout this article. Overall, their capacity for clinical application should not be overshadowed by their possible downsides, but taken as a reason to further research these intricate pathways and interactions. This is especially important today, when cardiovascular disease is a leading killer. Having therapeutics that can detect gene-level changes could be powerful disease prevention, meaning miRNAs could one day shift from their roles as midfielders to strikers, and ultimately transform cardiovascular care as we know it.

## Figures and Tables

**Figure 1 ijms-24-16207-f001:**
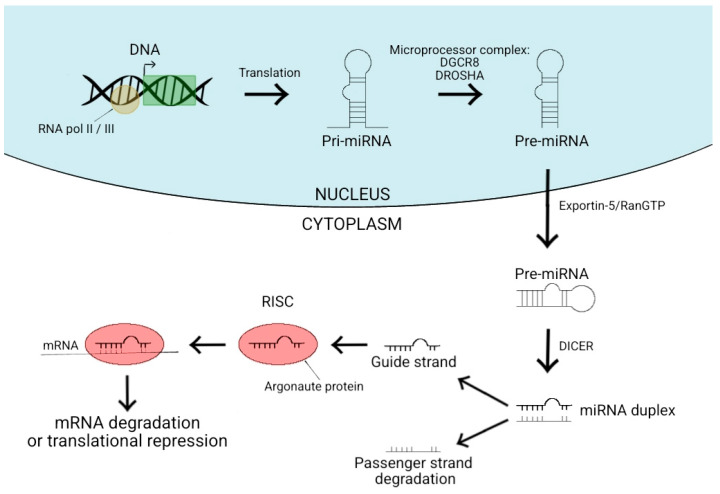
This is a diagram modified from Figure 1 in [8], depicting canonical miRNA biogenesis, and miRNA effects on mRNA.

**Figure 2 ijms-24-16207-f002:**
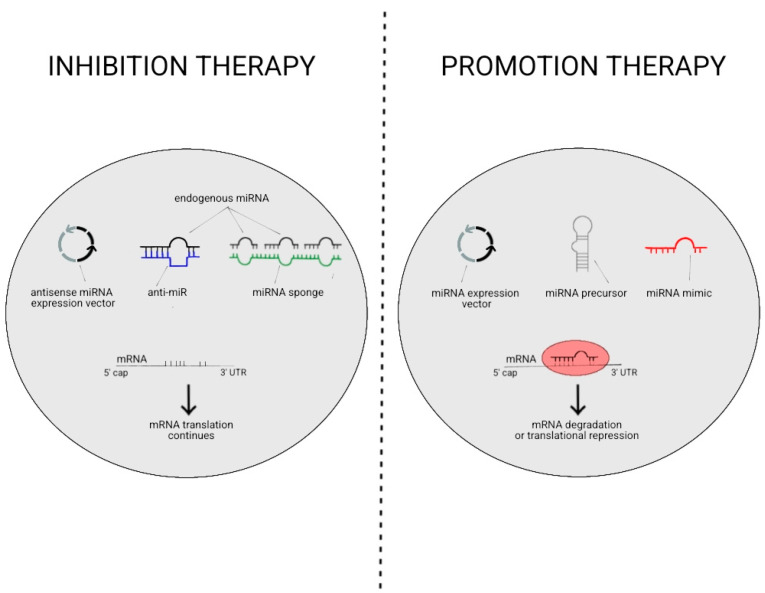
This is a diagram modified from Figure 3 in [59] and Figure 3 in [60], depicting therapeutic approaches to inhibit or promote miRNA activity.

**Figure 4 ijms-24-16207-f004:**
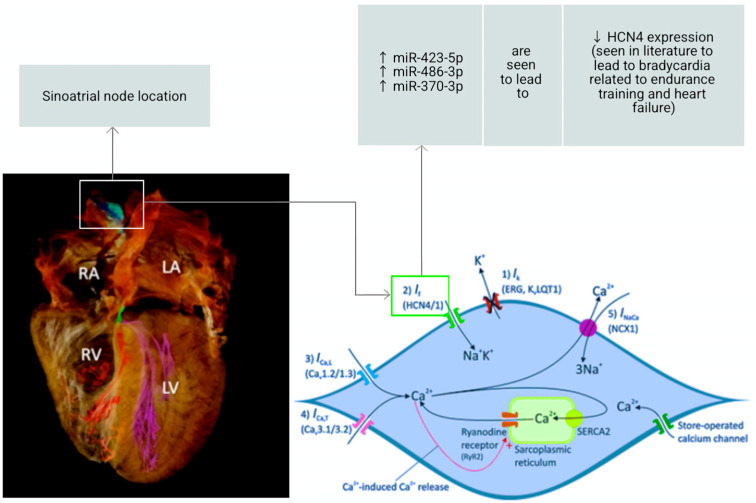
This is a diagram modified from Figure 7 in [96] and Figure 6 in [92]. It includes factors pertaining to *HCN4* downregulation in bradycardia (discussed in Section 2.2.2 and Section 2.2.3), highlighting three important miRNAs that are confirmed to target the *HCN4* channel [31,37,38] Arrows facing up depict upregulation of the key miRNAs while the arrow facing down reflects downregulation of *HCN4* expression.

**Table 1 ijms-24-16207-t001:** Summarises the miRNAs mentioned as biomarkers/therapeutics within the review.

Disease	Relevant miRNAs	Role as Biomarkers/Therapeutics	References
Coronary Artery Disease (CAD)	miR-126	Predictive ability to diagnose Type 2 diabetes Positive correlation with markers for co-morbid CAD	[14]
	miR-21	Positive correlation with disease severity	[15]
	miR-155	Inhibition leads to decreased pro-inflammatory cytokines	[16]
	miR-765, miR-149, miR-424, miR-133, miR-181	Can be used to differentiate between CAD and non-CAD patients	[17]
Atrial Fibrillation (AF)	miR-1	Decreasing levels improves arrhythmogenic symptoms	[18]
	miR-328	Elevated levels seen in AF Overexpression results in increased susceptibility to AF	[19,20,21]
	miR-1, miR-21, miR-26, miR-29b, miR-31, miR-328	Shown to protect from AF in animal models	[22]
	miR-29b	Overexpression results in decreased fibroblast	[23]
Heart Failure (HF)	miR-29	Protects against fibrosis	[24]
	miR-200a-3p	Inhibition can target several mediators of cardiac hypertrophy	[25,26]
	miR-378	Maintenance via genetic overexpression can decrease cardiac hypertrophy	[27]
	miR-423-5p	Elevated in HF patients Correlates with current HF markers Differential potential for HF Increased levels seen to correlate with better prognosis	[28,29]
	miR-132-3p	Repressed by antisense drug CDR132L	[5,30]
Sick Sinus Syndrome/Sinus Node Dysfunction (SSS/SND)	miR-486-3p	Confirmed to target HCN channels Depresses SAN automaticity Represses calcium channels Elevated in SAN tissue from HF patients Seen to alleviate bradycardia in COVID-19 patients	[31,32,33]
	miR-1 and miR-133	Those with AF show high HCN2/HCN4 and low miR-1/miR-133 compared to older patients with sinus rhythm	[34]
	miR-17-92 and miR-106b-25	Both can be targeted via Pitx2 (which positively regulates them)	[35]
	miR-1976	Seen to be increased in age-related SSS patient plasma	[36]
	miR-423-5p	Suggested to be increased in athletes Elevated in bradycardia related endurance training	[37]
	miR-370-3p	Decreasing in HF mice restores HCN4 mRNA while partially restoring ventricular function and decreasing mortality	[38]
	miR-448	Decreasing has anti-arrhythmic effects	[39]

## Data Availability

Data are contained within the article.

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
