# Peer review of "MicroRNAs: Midfielders of Cardiac Health, Disease and Treatment"

_ijms, 2023, doi:10.3390/ijms242216207_

Round 1
Reviewer 1 Report
Comments and Suggestions for Authors
A review by Asjad et al. is devoted to the role of microRNAs in cardiac health and disease, one of the hot topics in the fundamental cardiology. The main problem with the review is that it tries to cover an extremely broad topic and therefore, in my opinion, does it in a superficial way.
· What is the novelty of the review comparing to much more detailed works devoted to the involvement of miRNAs in the pathogenesis of coronary artery disease, atrial fibrillation, and heart failure separately (see doi:10.3389/fcvm.2021.632392, doi: 10.1016/j.pbiomolbio.2020.09.007, and doi:10.1161/CIRCULATIONAHA.119.042474 as the example)?
· Section 2.1 should be better placed after sections devoted to individual diseases. Some abstract data about miRNA-based therapeutics are ok, but again why data on miRNA biomarkers are so superficial and concentrated on extremely nonspecific myomiRs, when there are large panels of candidate miRNAs widely discussed in the literature?
· Authors often mention in the text or on the pictures quite big (but unfortunately still very far from full) sets of miRNAs related to some process and then describe just some of them (see lines 93-94 and Figure 3 as the example). What is the selection criterion? Why these miRNAs and not others?
· Table 1 is completely useless; the manuscript lacks summarizing tables showing the involvement of miRNAs in different pathological processes of the cardiovascular system.
Comments on the Quality of English Language
English is ok.
Reviewer 2 Report
Comments and Suggestions for Authors
The Authors submitted a Review centered on MicroRNAs and focusing on the role of miRNAs in the healthy and diseased heart, as well as their possible applications as therapeutics to treat cardiac pathologies. Despite the manuscript is well written and presented, I suggest to refine the latter specifying in the abstract and the text the a specifying the readers the Authors expect to hook with this review. In addition, I suggest to split into separate paragraph "conclusions" and "future perspectives". A Table that summarizes the most relevan litterature on possible applications as therapeutics of miRNAs to treat cardiac pathologies is also warranted.
Comments on the Quality of English LanguageMore attention should be paid to English grammar and structure. I also suggest to do a double check for typos.
Reviewer 3 Report
Comments and Suggestions for Authors
We read with great interest the article by Emman Asjad and Halina Dobrzynski where they describe the functions of miRNA in the area of Cardiac Health and Disease
Comments:
The article is well-written and has excellent information, I would suggest that the authors expand Tabe 1 to describe the functions of the different genes and how they are related to cardiac disease and please add references.
Second, the different miRNAs described need to be inserted in a table to describe their functions and what are the downstream targets.
Third, the authors should have a separate section discussing miRNAs as therapeutic targets
Minor Comments:
the author list is wrong:
Emman Asjad 1*, Halina Dobrzynski 1* and 2????/
Round 2
Reviewer 1 Report
Comments and Suggestions for Authors
I believe the authors did not respond to my comments in the appropriate manner. The review still seems to be extremely superficial due to the choice of a very broad topic for discussion. Authors claim that the novelty of their work is in the description of the involvement of miRNAs in age-related sick sinus syndrome; sinus node dysfunction linked to atrial fibrillation, heart failure and bradycardia; key miRNAs involved in heart development and heart rate modulation. However a fast search highlights a review [10.1016/j.carpath.2020.107296], which gives much deeper understanding of miRNAs involvement in heart development, and a review [10.1016/j.pbiomolbio.2022.11.005], which describes in detail the role of miRNAs in SAN activity in health and disease including SSS. Several microRNAs present in the illustrations are still not discussed in the text and vice versa without any reasons of it mentioned. It seems more like some sort of cherry picking without any general idea.
Author Response
We believe that we have addressed your comments in an appropriate manner. We highlighted the novelty of our unique review and guided potential readers to key references that discuss this topic in more details. We thank you for your suggestion but there is nothing else we can add.
Reviewer 2 Report
Comments and Suggestions for Authors
The Authors addressed all the comments raised by the Reviewers. Therefore, the manuscript significantly improved in its content. I have no further comments or edits.
Author Response
Thank you for your kind comments.